# HUMAN MOTION DIFFUSION MODEL

**Guy Tevet, Sigal Raab, Brian Gordon, Yonatan Shafir,**
**Daniel Cohen-Or and Amit H. Bermano**
Tel Aviv University, Israel
`guytevet@mail.tau.ac.il`

## ABSTRACT

Natural and expressive human motion generation is the holy grail of computer animation. It is a challenging task, due to the diversity of possible motion, human perceptual sensitivity to it, and the difficulty of accurately describing it. Therefore, current generative solutions are either low-quality or limited in expressiveness. Diffusion models, which have already shown remarkable generative capabilities in other domains, are promising candidates for human motion due to their many-to-many nature, but they tend to be resource hungry and hard to control. In this paper, we introduce Motion Diffusion Model (MDM), a carefully adapted classifier-free diffusion-based generative model for the human motion domain. MDM is transformer-based, combining insights from motion generation literature. A notable design-choice is the prediction of the sample, rather than the noise, in each diffusion step. This facilitates the use of established geometric losses on the locations and velocities of the motion, such as the foot contact loss. As we demonstrate, MDM is a generic approach, enabling different modes of conditioning, and different generation tasks. We show that our model is trained with lightweight resources and yet achieves state-of-the-art results on leading benchmarks for text-to-motion and action-to-motion [1]. `https://guytevet.github.io/mdm-page/`.

## 1 INTRODUCTION

Human motion generation is a fundamental task in computer animation, with applications spanning from gaming to robotics. It is a challenging field, due to several reasons, including the vast span of possible motions, and the difficulty and cost of acquiring high quality data. For the recently emerging text-to-motion setting, where motion is generated from natural language, another inherent problem is data labeling. For example, the label "kick" could refer to a soccer kick, as well as a Karate one. At the same time, given a specific kick there are many ways to describe it, from how it is performed to the emotions it conveys, constituting a many-to-many problem. Current approaches have shown success in the field, demonstrating plausible mapping from text to motion (Petrovich et al., 2022; Tevet et al., 2022; Ahuja & Morency, 2019). All these approaches, however, still limit the learned distribution since they mainly employ auto-encoders or VAEs (Kingma & Welling, 2013) (implying a one-to-one mapping or a normal latent distribution respectively). In this aspect, diffusion models are a better candidate for human motion generation, as they are free from assumptions on the target distribution, and are known for expressing well the many-to-many distribution matching problem we have described.

Diffusion models (Sohl-Dickstein et al., 2015; Song & Ermon, 2020; Ho et al., 2020) are a generative approach that is gaining significant attention in the computer vision and graphics community. When trained for conditioned generation, recent diffusion models (Ramesh et al., 2022; Saharia et al., 2022b) have shown breakthroughs in terms of image quality and semantics. The competence of these models have also been shown for other domains, including videos (Ho et al., 2022), and 3D point clouds (Luo & Hu, 2021). The problem with such models, however, is that they are notoriously resource demanding and challenging to control.

In this paper, we introduce Motion Diffusion Model (MDM) — a carefully adapted diffusion based generative model for the human motion domain. Being diffusion-based, MDM gains from the na-

---

[1]Code can be found at `https://github.com/GuyTevet/motion-diffusion-model`.

"A person kicks with their left leg."

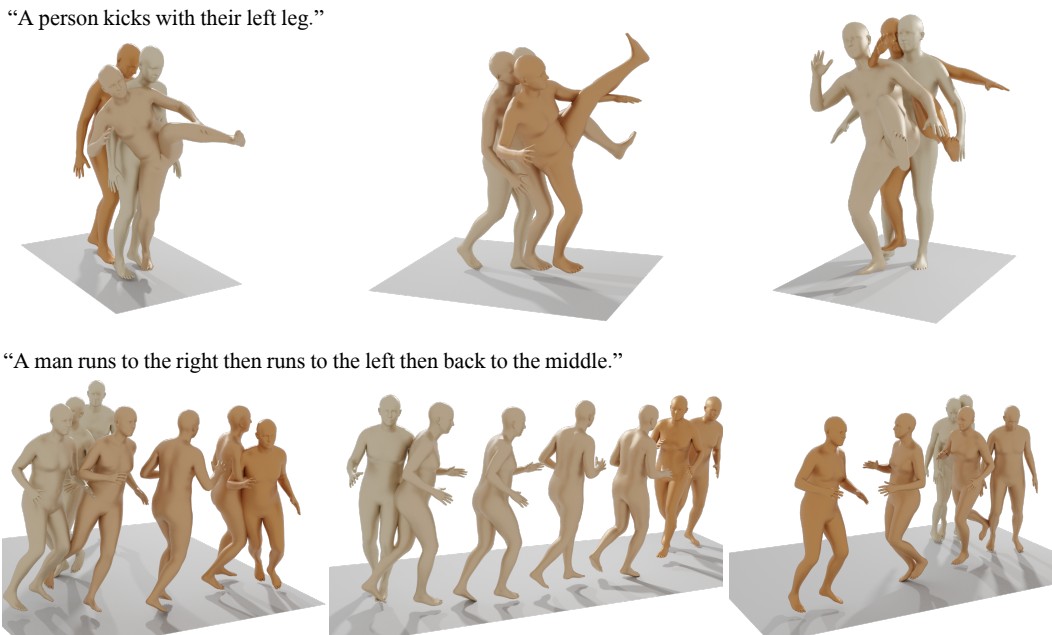

"A man runs to the right then runs to the left then back to the middle."

Figure 1: Our Motion Diffusion Model (MDM) reflects the many-to-many nature of text-to-motion mapping by generating diverse motions given a text prompt. Our custom architecture and geometric losses help yielding high-quality motion. Darker color indicates later frames in the sequence.

tive aforementioned many-to-many expression of the domain, as evidenced by the resulting motion quality and diversity (Figure 1). In addition, MDM combines insights already well established in the motion generation domain, helping it be significantly more lightweight and controllable.

First, instead of the ubiquitous U-net (Ronneberger et al., 2015) backbone, MDM is transformer-based. As we demonstrate, our architecture (Figure 2) is lightweight and better fits the temporal and spatially irregular nature of motion data (represented as a collection of joints). A large volume of motion generation research is devoted to learning using geometric losses (Kocabas et al., 2020; Harvey et al., 2020; Aberman et al., 2020). Some, for example, regulate the velocity of the motion (Petrovich et al., 2021) to prevent jitter, or specifically consider foot sliding using dedicated terms (Shi et al., 2020). Consistently with these works, we show that applying geometric losses in the diffusion setting improves generation.

The MDM framework has a generic design enabling different forms of conditioning. We showcase three tasks: text-to-motion, action-to-motion, and unconditioned generation. We train the model in a classifier-free manner (Ho & Salimans, 2022), which enables trading-off diversity to fidelity, and sampling both conditionally and unconditionally from the same model. In the text-to-motion task, our model generates coherent motions (Figure 1) that achieve state-of-the-art results on the HumanML3D (Guo et al., 2022a) and KIT (Plappert et al., 2016) benchmarks. Moreover, our user study shows that human evaluators prefer our generated motions over real motions 42% of the time (Figure 4(a)). In action-to-motion, MDM outperforms the state-of-the-art (Guo et al., 2020; Petrovich et al., 2021), even though they were specifically designed for this task, on the common HumanAct12 (Guo et al., 2020) and UESTC (Ji et al., 2018) benchmarks.

Lastly, we also demonstrate completion and editing. By adapting diffusion image-inpainting (Song et al., 2020b; Saharia et al., 2022a), we set a motion prefix and suffix, and use our model to fill in the gap. Doing so under a textual condition guides MDM to fill the gap with a specific motion that still maintains the semantics of the original input. By performing inpainting in the joints space rather than temporally, we also demonstrate the semantic editing of specific body parts, without changing the others (Figure 3).

Overall, we introduce Motion Diffusion Model, a motion framework that achieves state-of-the-art quality in several motion generation tasks, while requiring only about three days of training on a single mid-range GPU. It supports geometric losses, which are non trivial to the diffusion setting,

but are crucial to the motion domain, and offers the combination of state-of-the-art generative power with well thought-out domain knowledge.

## 2 RELATED WORK

### 2.1 HUMAN MOTION GENERATION

Neural motion generation, learned from motion capture data, can be conditioned by any signal that describes the motion. Many works use parts of the motion itself for guidance. Some predict motion from its prefix poses (Fragkiadaki et al., 2015; Martinez et al., 2017; Hernandez et al., 2019; Guo et al., 2022b). Others (Harvey & Pal, 2018; Kaufmann et al., 2020; Harvey et al., 2020; Duan et al., 2021) solve in-betweening and super-resolution tasks using bi-directional GRU (Cho et al., 2014) and Transformer (Vaswani et al., 2017) architectures. Holden et al. (2016) use auto-encoder to learn motion latent representation, then utilize it to edit and control motion with spatial constraints such as root trajectory and bone lengths. Motion can be controlled with a high-level guidance given from action class (Guo et al., 2020; Petrovich et al., 2021; Cervantes et al., 2022), audio (Li et al., 2021; Aristidou et al., 2022) and natural language (Ahuja & Morency, 2019; Petrovich et al., 2022). In most cases authors suggests a dedicated approach to map each conditioning domain into motion.

In recent years, the leading approach for the *Text-to-Motion* task is to learn a shared latent space for language and motion. JL2P (Ahuja & Morency, 2019) learns the KIT motion-language dataset (Plappert et al., 2016) with an auto-encoder, limiting one-to-one mapping from text to motion. TEMOS (Petrovich et al., 2022) and T2M (Guo et al., 2022a) suggest using a VAE (Kingma & Welling, 2013) to map a text prompt into a normal distribution in latent space. Recently, MotionCLIP (Tevet et al., 2022) leverages the shared text-image latent space learned by CLIP (Radford et al., 2021) to expand text-to-motion out of the data limitations and enabled latent space editing.

The human motion manifold can also be learned without labels, as shown by Holden et al. (2016), V-Poser (Pavlakos et al., 2019), and more recently the dedicated MoDi architecture (Raab et al., 2022). We show that our model is capable for such an unsupervised setting as well.

### 2.2 DIFFUSION GENERATIVE MODELS

Diffusion models (Sohl-Dickstein et al., 2015; Song & Ermon, 2020) are a class of neural generative models, based on the stochastic diffusion process as it is modeled in Thermodynamics. In this setting, a sample from the data distribution is gradually noised by the diffusion process. Then, a neural model learns the *reverse process* of gradually denoising the sample. Sampling the learned data distribution is done by denoising a pure initial noise. Ho et al. (2020) and Song et al. (2020a) further developed the practices for image generation applications. For conditioned generation, Dhariwal & Nichol (2021), introduced classifier-guided diffusion, which was later on adapted by GLIDE (Nichol et al., 2021) to enable conditioning over CLIP textual representations. The Classifier-Free Guidance approach Ho & Salimans (2022) enables conditioning while trading-off fidelity and diversity, and achieves better results (Nichol et al., 2021). In this paper, we implement text-to-motion by conditioning on CLIP in a classifier-free manner, similarly to text-to-image (Ramesh et al., 2022; Saharia et al., 2022b). Local editing of images is typically defined as an inpainting problem, where a part of the image is constant, and the inpainted part is denoised by the model, possibly under some condition (Song et al., 2020b; Saharia et al., 2022a). We adapt this technique to edit motion's specific body parts or temporal intervals (in-betweening) according to an optional condition.

Closer to our context, Gu et al. (2022) used the diffusion formulation to model the stochasticity of human trajectory prediction. More recently, concurrent to this work, Zhang et al. (2022) and Kim et al. (2022) have suggested diffusion models for motion generation. Our work requires significantly fewer GPU resources and makes design choices that enable geometric losses, which improve results.

## 3 MOTION DIFFUSION MODEL

An overview of our method is described in Figure 2. Our goal is to synthesize a human motion $x^{1:N}$ of length $N$ given an arbitrary condition $c$. This condition can be any real-world signal that will dictate the synthesis, such as audio (Li et al., 2021; Aristidou et al., 2022), natural language (text-to-motion) (Tevet et al., 2022; Guo et al., 2022a) or a discrete class (action-to-motion) (Guo et al., 2020; Petrovich et al., 2021). In addition, unconditioned motion generation is also possible, which we denote as the null condition $c = \emptyset$. The generated motion $x^{1:N} = \{x^i\}_{i=1}^{N}$ is a sequences

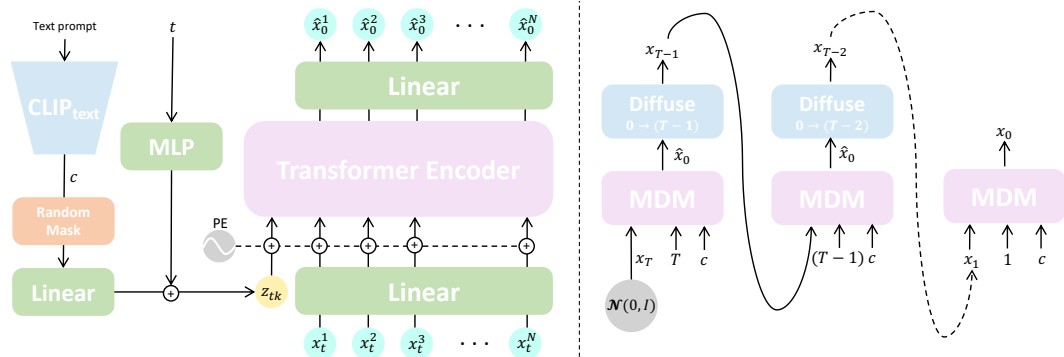

Figure 2: **(Left) Motion Diffusion Model (MDM) overview.** The model is fed a motion sequence $x_t^{1:N}$ of length $N$ in a noising step $t$, as well as $t$ itself and a conditioning code $c$. $c$, a CLIP (Radford et al., 2021) based textual embedding in this case, is first randomly masked for classifier-free learning and then projected together with $t$ into the input token $z_{tk}$. In each sampling step, the transformer-encoder predicts the final clean motion $\hat{x}_0^{1:N}$. **(Right) Sampling MDM.** Given a condition $c$, we sample random noise $x_T$ at the dimensions of the desired motion, then iterate from $T$ to $1$. At each step $t$, MDM predicts the clean sample $\hat{x}_0$, and diffuses it back to $x_{t-1}$.

of human poses represented by either joint rotations or positions $x^i \in \mathbb{R}^{J \times D}$, where $J$ is the number of joints and $D$ is the dimension of the joint representation. MDM can accept motion represented by either locations, rotations, or both (see Section 4).

**Framework.** Diffusion is modeled as a Markov noising process, $\{x_t^{1:N}\}_{t=0}^T$, where $x_0^{1:N}$ is drawn from the data distribution and

$$q(x_t^{1:N}|x_{t-1}^{1:N}) = \mathcal{N}(\sqrt{\alpha_t}x_{t-1}^{1:N}, (1-\alpha_t)I), \tag{1}$$

where $\alpha_t \in (0,1)$ are constant hyper-parameters. When $\alpha_t$ is small enough, we can approximate $x_T^{1:N} \sim \mathcal{N}(0,I)$. From here on we use $x_t$ to denote the full sequence at noising step $t$.

In our context, conditioned motion synthesis models the distribution $p(x_0|c)$ as the reversed diffusion process of gradually cleaning $x_T$. Instead of predicting $\epsilon_t$ as formulated by Ho et al. (2020), we follow Ramesh et al. (2022) and use an equivalent formulation to predict the signal itself, i.e., $\hat{x}_0 = G(x_t, t, c)$ with the *simple* objective (Ho et al., 2020),

$$\mathcal{L}_{\text{simple}} = E_{x_0 \sim q(x_0|c), t \sim [1,T]}[\|x_0 - G(x_t, t, c)\|_2^2] \tag{2}$$

**Geometric losses.** In the motion domain, generative networks are standardly regularized using geometric losses Petrovich et al. (2021); Shi et al. (2020). These losses enforce physical properties and prevent artifacts, encouraging natural and coherent motion. In this work we experiment with three common geometric losses that regulate (1) positions (in case we predict rotations), (2) foot contact, and (3) velocities.

$$\mathcal{L}_{\text{pos}} = \frac{1}{N} \sum_{i=1}^N \|FK(x_0^i) - FK(\hat{x}_0^i)\|_2^2, \tag{3}$$

$$\mathcal{L}_{\text{foot}} = \frac{1}{N-1} \sum_{i=1}^{N-1} \|(FK(\hat{x}_0^{i+1}) - FK(\hat{x}_0^i)) \cdot f_i\|_2^2, \tag{4}$$

$$\mathcal{L}_{\text{vel}} = \frac{1}{N-1} \sum_{i=1}^{N-1} \|(x_0^{i+1} - x_0^i) - (\hat{x}_0^{i+1} - \hat{x}_0^i)\|_2^2 \tag{5}$$

In case we predict joint rotations, $FK(\cdot)$ denotes the forward kinematic function converting joint rotations into joint positions (otherwise, it denotes the identity function). $f_i \in \{0,1\}^J$ is the binary foot contact mask for each frame $i$. Relevant only to feet, it indicates whether they touch the ground, and are set according to binary ground truth data (Shi et al., 2020). In essence, it mitigates the foot-sliding effect by nullifying velocities when touching the ground. Overall, our training loss is $\mathcal{L} = \mathcal{L}_{\text{simple}} + \lambda_{\text{pos}}\mathcal{L}_{\text{pos}} + \lambda_{\text{vel}}\mathcal{L}_{\text{vel}} + \lambda_{\text{foot}}\mathcal{L}_{\text{foot}}$.

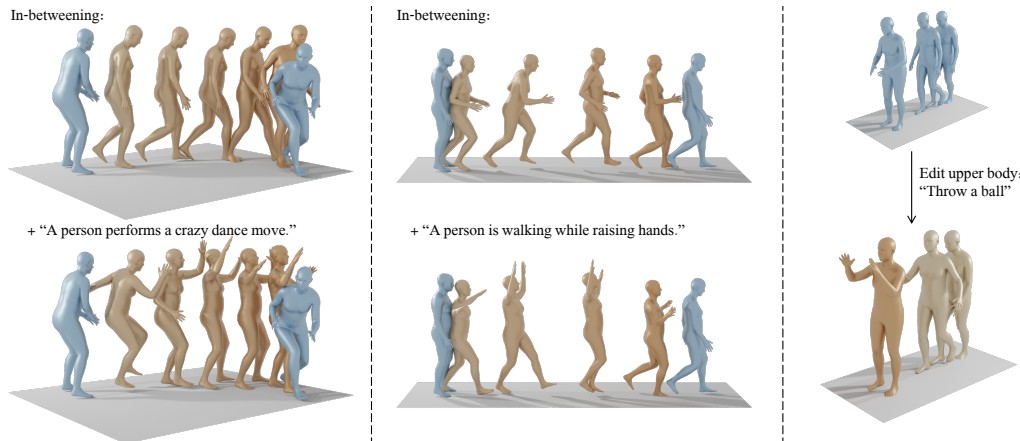

In-betweening:

+ "A person performs a crazy dance move."

In-betweening:

+ "A person is walking while raising hands."

Edit upper body: "Throw a ball"

Figure 3: **Editing applications.** Light blue frames represent motion input and bronze frames are the generated motion. Motion in-betweening (left+center) can be performed conditioned on text or without condition by the same model. Specific body part editing using text is demonstrated on the right: the lower body joints are fixed to the input motion while the upper body is altered to fit the input text prompt.

**Model.** Our model is illustrated in Figure 2. We implement $G$ with a straightforward transformer (Vaswani et al., 2017) encoder-only architecture. The transformer architecture is temporally aware, enabling learning and generating variable-length motions, and is well-proven for the motion domain (Petrovich et al., 2021; Duan et al., 2021; Aksan et al., 2021). The noise time-step $t$ and the condition code $c$ are each projected to the transformer dimension by separate feed-forward networks, then summed to yield the token $z_{tk}$. Each frame of the noised input $x_t$ is linearly projected into the transformer dimension and summed with a standard positional embedding. $z_{tk}$ and the projected frames are then fed to the encoder. Excluding the first output token (corresponding to $z_{tk}$), the encoder result is projected back to the original motion dimensions, and serves as the prediction $\hat{x}_0$. We implement text-to-motion by encoding the text prompt to $c$ with CLIP (Radford et al., 2021) text encoder, and action-to-motion with learned embeddings per class.

**Sampling** from $p(x_0|c)$ is done in an iterative manner, according to Ho et al. (2020). In every time step $t$ we predict the clean sample $\hat{x}_0 = G(x_t, t, c)$ and noise it back to $x_{t-1}$. This is repeated from $t = T$ until $x_0$ is achieved (Figure 2 right). We train our model $G$ using classifier-free guidance (Ho & Salimans, 2022). In practice, $G$ learns both the conditioned and the unconditioned distributions by randomly setting $c = \emptyset$ for 10% of the samples, such that $G(x_t, t, \emptyset)$ approximates $p(x_0)$. Then, when sampling $G$ we can trade-off diversity and fidelity by interpolating or even extrapolating the two variants using $s$: $G_s(x_t, t, c) = G(x_t, t, \emptyset) + s \cdot (G(x_t, t, c) - G(x_t, t, \emptyset))$.

**Editing.** We enable motion in-betweening in the temporal domain, and body part editing in the spatial domain, by adapting diffusion inpainting to motion data. Editing is done only during sampling, without any training involved. Given a subset of the motion sequence inputs, when sampling the model (Figure 2 right), at each iteration we overwrite $\hat{x}_0$ with the input part of the motion. This encourages the generation to remain coherent to original input, while completing the missing parts. In the temporal setting, the prefix and suffix frames of the motion sequence are the input, and we solve a motion in-betweening problem (Harvey et al., 2020). Editing can be done either conditionally or unconditionally (by setting $c = \emptyset$). In the spatial setting, we show that body parts can be re-synthesized according to a condition $c$ while keeping the rest intact, through the use of the same completion technique.

## 4 EXPERIMENTS

We implement MDM for three motion generation tasks: Text-to-Motion(4.1), Action-to-Motion(4.2) and unconditioned generation(5.2. Each sub-section reviews the data and metrics of the used benchmarks, provides implementation details, and presents qualitative and quantitative results. Then, we show implementations of motion in-betweening (both conditioned and unconditioned) and body-part editing by adapting diffusion inpainting to motion (5.1). Our models have been trained with $T = 1000$ noising steps and a cosine noise schedule. In Appendix A.1, we experiment with differ-

ent values of $T$. All of them have been trained on a single *NVIDIA GeForce RTX 2080 Ti* GPU for a period of about 3 days.

## 4.1 TEXT-TO-MOTION

Text-to-motion is the task of generating motion given an input text prompt. The output motion is expected to be both implementing the textual description, and a valid sample from the data distribution (i.e. adhering to general human abilities and the rules of physics). In addition, for each text prompt, we also expect a distribution of motions matching it, rather than just a single result. We evaluate our model using two leading benchmarks - KIT (Plappert et al., 2016) and HumanML3D (Guo et al., 2022a), over the set of metrics suggested by Guo et al. (2022a): *R-precision* and *Multimodal-Dist* measure the relevancy of the generated motions to the input prompts, *FID* measures the dissimilarity between the generated and ground truth distributions (in latent space), *Diversity* measures the variability in the resulting motion distribution, and *MultiModality* is the average variance given a single text prompt. For the full implementation of the metrics, please refer to Guo et al. (2022a). We use HumanML3D as a platform to compare different backbones of our model, discovering that the diffusion framework is relatively agnostic to this attribute. In addition, we conduct a user study comparing our model to current art and ground truth motions.

**Data.** HumanML3D is a recent dataset, textually re-annotating motion capture from the AMASS (Mahmood et al., 2019) and HumanAct12 (Guo et al., 2020) collections. It contains $14,616$ motions annotated by $44,970$ textual descriptions. In addition, it suggests a redundant data representation including a concatenation of root velocity, joint positions, joint velocities, joint rotations and the foot contact binary labels. We also use in this section the same representation for the KIT dataset, brought by the same publishers. Although limited in the number ($3,911$) and the diversity of samples, most of the text-to-motion research is based on KIT, hence we view it as important to evaluate using it as well.

**Implementation.** In addition to our Transformer encoder-only backbone (Section 3), we experiment MDM with three more backbones: (1) *Transformer decoder* injects $z_{tk}$ through the cross-attention layer, instead of as an input token. (2) *Transformer decoder + input token*, where $z_{tk}$ is injected both ways, (3) *GRU* (Cho et al., 2014) concatenate $z_{tk}$ to each input frame , and (4) a U-net adaptation for motion data (Table 1). U-net is adapted to motion by replacing the 2D convolution filters with 1D convolutions in the temporal axis, such that joints are considered as channels. This is due to the irregular behavior of the joint axis. Our models were trained with batch size $64$, $8$ layers (except GRU that was optimal at $2$), and latent dimension $512$. To encode the text we use a frozen *CLIP-ViT-B/32* model. In addition, we experiment with replacing CLIP with sentence-BERT (Reimers & Gurevych, 2019) - a BERT-based (Devlin et al., 2019) text encoder. The full details can be found in our published code[1] and Appendix C. Each model was trained for $500K$ steps, after which a checkpoint was chosen that minimizes the FID metric to be reported. Since foot contact and joint locations are explicitly represented in HumanML3D, we don't apply geometric losses in this section. We evaluate our models with guidance-scale $s = 2.5$ which provides a diversity-fidelity sweet spot (Figure 4).

**Quantitative evaluation.** We evaluate and compare our models to current art (JL2P Ahuja & Morency (2019), Text2Gesture (Bhattacharya et al., 2021), and T2M (Guo et al., 2022a)) with the metrics suggested by Guo et al. (2022a). As can be seen, MDM achieves state-of-the-art results in *FID*, *Diversity*, and *MultiModality*, indicating high diversity per input text prompt, and high-quality samples, as can also be seen qualitatively in Figure 1.

**User study.** We asked 31 users to choose between MDM and state-of-the-art works in a side-by-side view, with both samples generated from the same text prompt randomly sampled from the KIT test set. We repeated this process with 10 samples per model and 10 repetitions per sample. This user study enabled a comparison with the recent TEMOS model (Petrovich et al., 2022), which was not included in the HumanML3D benchmark. Fig. 4 shows that most of the time, MDM was preferred over the compared models, and even preferred over ground truth samples in $42.3\%$ of the cases.

## 4.2 ACTION-TO-MOTION

Action-to-motion is the task of generating motion given an input action class, represented by a scalar. The output motion should faithfully animate the input action, and at the same time be natural and reflect the distribution of the dataset on which the model is trained. Two dataset are commonly used to evaluate action-to-motion models: HumanAct12 (Guo et al., 2020) and UESTC (Ji et al., 2018).

| Method | R Precision (top 3)↑ | FID↓ | Multimodal Dist↓ | Diversity→ | Multimodality↑ |
|---|---|---|---|---|---|
| Real | $0.797^{\pm.002}$ | $0.002^{\pm.000}$ | $2.974^{\pm.008}$ | $9.503^{\pm.065}$ | - |
| JL2P | $0.486^{\pm.002}$ | $11.02^{\pm.046}$ | $5.296^{\pm.008}$ | $7.676^{\pm.058}$ | - |
| Text2Gesture | $0.345^{\pm.002}$ | $7.664^{\pm.030}$ | $6.030^{\pm.008}$ | $6.409^{\pm.071}$ | - |
| T2M | $\mathbf{0.740}^{\pm.003}$ | $1.067^{\pm.002}$ | $\mathbf{3.340}^{\pm.008}$ | $9.188^{\pm.002}$ | $2.090^{\pm.083}$ |
| MDM (ours) | $0.611^{\pm.007}$ | $\mathbf{0.544}^{\pm.044}$ | $5.566^{\pm.027}$ | $\mathbf{9.559}^{\pm.086}$ | $2.799^{\pm.072}$ |
| + sent-BERT | $0.609^{\pm.006}$ | $0.586^{\pm.036}$ | $5.504^{\pm.03}$ | $9.666^{\pm.095}$ | $2.707^{\pm.188}$ |
| MDM (decoder) | $0.608^{\pm.005}$ | $0.767^{\pm.085}$ | $5.507^{\pm.020}$ | $9.176^{\pm.070}$ | $\mathbf{2.927}^{\pm.125}$ |
| + input token | $0.621^{\pm.005}$ | $0.567^{\pm.051}$ | $5.424^{\pm.022}$ | $9.425^{\pm.060}$ | $2.834^{\pm.095}$ |
| MDM (U-net) | $0.603^{\pm.006}$ | $1.137^{\pm.008}$ | $5.629^{\pm.032}$ | $8.958^{\pm.098}$ | $2.636^{\pm.214}$ |
| MDM (GRU) | $0.645^{\pm.005}$ | $4.569^{\pm.150}$ | $5.325^{\pm.026}$ | $7.688^{\pm.082}$ | $1.2646^{\pm.024}$ |

Table 1: **Quantitative results on the HumanML3D test set.** All methods use the real motion length from the ground truth. '→' means results are better if the metric is closer to the real distribution. We run all the evaluation 20 times (except *MultiModality* runs 5 times) and ± indicates the 95% confidence interval. **Bold** indicates best result.

| Method | R Precision (top 3)↑ | FID↓ | Multimodal Dist↓ | Diversity→ | Multimodality↑ |
|---|---|---|---|---|---|
| Real | $0.779^{\pm.006}$ | $0.031^{\pm.004}$ | $2.788^{\pm.012}$ | $11.08^{\pm.097}$ | - |
| JL2P | $0.483^{\pm.005}$ | $6.545^{\pm.072}$ | $5.147^{\pm.030}$ | $9.073^{\pm.100}$ | - |
| Text2Gesture | $0.338^{\pm.005}$ | $12.12^{\pm.183}$ | $6.964^{\pm.029}$ | $9.334^{\pm.079}$ | - |
| T2M | $\mathbf{0.693}^{\pm.007}$ | $2.770^{\pm.109}$ | $\mathbf{3.401}^{\pm.008}$ | $\mathbf{10.91}^{\pm.119}$ | $1.482^{\pm.065}$ |
| MDM (ours) | $0.396^{\pm.004}$ | $\mathbf{0.497}^{\pm.021}$ | $9.191^{\pm.022}$ | $10.847^{\pm.109}$ | $\mathbf{1.907}^{\pm.214}$ |

Table 2: **Quantitative results on the KIT test set.**

We evaluate our model using the set of metrics suggested by Guo et al. (2020), namely Fréchet Inception Distance (FID), action recognition accuracy, diversity and multimodality. The combination of these metrics makes a good measure of the realism and diversity of generated motions.

**Data.** HumanAct12 (Guo et al., 2020) offers approximately 1200 motion clips, organized into 12 action categories, with 47 to 218 samples per label. UESTC (Ji et al., 2018) consists of 40 action classes, 40 subjects and 25K samples, and is split to train and test. We adhere to the cross-subject testing protocol used by current works, with 225-345 samples per action class. For both datasets we use the sequences provided by Petrovich et al. (2021).

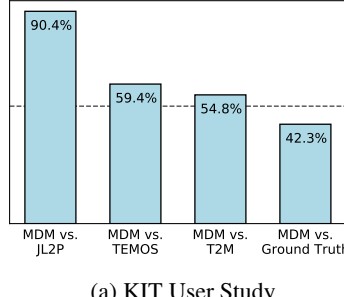

(a) KIT User Study

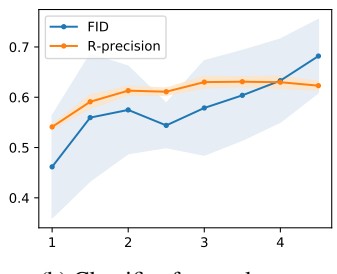

(b) Classifier-free scale sweep

Figure 4: **(a) Text-to-motion user study for the KIT dataset**. Each bar represents the preference rate of MDM over the compared model. MDM was preferred over the other models in most of the time, and 42.3% of the cases even over ground truth samples. The dashed line marks 50%. **(b) Guidance-scale sweep for HumanML3D dataset.** *FID* (lower is better) and *R-precision* (higher is better) metrics as a function of the scale $s$, draws an accuracy-fidelity sweet spot around $s = 2.5$.

| Method | FID$\downarrow$ | Accuracy$\uparrow$ | Diversity$\rightarrow$ | Multimodality$\rightarrow$ |
|---|---|---|---|---|
| Real (INR) | $0.020^{\pm.010}$ | $0.997^{\pm.001}$ | $6.850^{\pm.050}$ | $2.450^{\pm.040}$ |
| Real (ours) | $0.050^{\pm.000}$ | $0.990^{\pm.000}$ | $6.880^{\pm.020}$ | $2.590^{\pm.010}$ |
| Action2Motion (2020) | $0.338^{\pm.015}$ | $0.917^{\pm.003}$ | $6.879^{\pm.066}$ | $\underline{2.511}^{\pm.023}$ |
| ACTOR (2021) | $0.120^{\pm.000}$ | $0.955^{\pm.008}$ | $\mathbf{6.840}^{\pm.030}$ | $2.530^{\pm.020}$ |
| INR (2022) | $\underline{0.088}^{\pm.004}$ | $\underline{0.973}^{\pm.001}$ | $6.881^{\pm.048}$ | $2.569^{\pm.040}$ |
| MDM (ours) | $0.100^{\pm.000}$ | $\mathbf{0.990}^{\pm.000}$ | $\underline{6.860}^{\pm.050}$ | $2.520^{\pm.010}$ |
| w/o foot contact | $\mathbf{0.080}^{\pm.000}$ | $\mathbf{0.990}^{\pm.000}$ | $6.810^{\pm.010}$ | $\mathbf{2.580}^{\pm.010}$ |

Table 3: **Evaluation of action-to-motion on the HumanAct12 dataset.** Our model leads the board in three out of four metrics. Ground-truth evaluation results are slightly different for each of the works, due to implementation differences, such as python package versions. It is important to assess the diversity and multimodality of each model using its own ground-truth results, as they are measured by their distance from GT. We show the GT metrics measured by our model and by the leading compared work, INR (Cervantes et al., 2022). **Bold** indicates best result, underline indicates second best, $\pm$ indicates 95% confidence interval, $\rightarrow$ indicates that closer to real is better.

| Method | $\text{FID}_{\text{train}}\downarrow$ | $\text{FID}_{\text{test}}\downarrow$ | Accuracy$\uparrow$ | Diversity$\rightarrow$ | Multimodality$\rightarrow$ |
|---|---|---|---|---|---|
| Real | $2.92^{\pm.26}$ | $2.79^{\pm.29}$ | $0.988^{\pm.001}$ | $33.34^{\pm.320}$ | $14.16^{\pm.06}$ |
| ACTOR (2021) | $20.49^{\pm2.31}$ | $23.43^{\pm2.20}$ | $0.911^{\pm.003}$ | $31.96^{\pm.33}$ | $14.52^{\pm.09}$ |
| INR (2022) (best variation) | $\mathbf{9.55}^{\pm.06}$ | $15.00^{\pm.09}$ | $0.941^{\pm.001}$ | $31.59^{\pm.19}$ | $14.68^{\pm.07}$ |
| MDM (ours) | $9.98^{\pm1.33}$ | $\mathbf{12.81}^{\pm1.46}$ | $\underline{0.950}^{\pm.000}$ | $\underline{33.02}^{\pm.28}$ | $\mathbf{14.26}^{\pm.12}$ |
| w/o foot contact | $\underline{9.69}^{\pm.81}$ | $\underline{13.08}^{\pm2.32}$ | $\mathbf{0.960}^{\pm.000}$ | $\mathbf{33.10}^{\pm.29}$ | $14.06^{\pm.05}$ |

Table 4: **Evaluation of action-to-motion on the UESTC dataset.** The performance improvement with our model shows a clear gap from state-of-the-art. **Bold** indicates best result, underline indicates second best, $\pm$ indicates 95% confidence interval, $\rightarrow$ indicates that closer to real is better.

**Implementation.** The implementation presented in Figure 2 holds for all the variations of our work. In the case of action-to-motion, the only change would be the substitution of the text embedding by an action embedding. Since action is represented by a scalar, its embedding is fairly simple; each input action class scalar is converted into a learned embedding of the transformer dimension.

The experiments have been run with batch size $64$, a latent dimension of $512$, and an encoder-transformer architecture. Training on HumanAct12 and UESTC has been carried out for $750K$ and $2M$ steps respectively. In our tables we display the evaluation of the checkpoint that minimizes the FID metric.

**Quantitative evaluation.** Tables 3 and 4 reflect MDM's performance on the HumanAct12 and UESTC datasets respectively. We conduct 20 evaluations, with 1000 samples in each, and report their average and a 95% confidence interval. We test two variations, with and without foot contact loss. Full ablation study for geometric losses is presented in Appendix A.2. Our model leads the board for both datasets. The variation with no foot contact loss attains slightly better results; nevertheless, as shown in our supplementary video, the contribution of foot contact loss to the quality of results is important, and without it we witness artifacts such as shakiness and unnatural gestures.

## 5 Additional Applications

### 5.1 Motion Editing

In this section we implement two motion editing applications - **in-betweening** and **body part editing**, both using the same approach in the temporal and spatial domains correspondingly. For **in-betweening**, we fix the first and last $25\%$ of the motion, leaving the model to generate the remaining $50\%$ in the middle. For **body part editing**, we fix the joints we don't want to edit and leave the

model to generate the rest. In particular, we experiment with editing the upper body joints only. In figure 3 we show that in both cases, using the method described in Section 3 generates smooth motions that adhere both to the fixed part of the motion and the condition (if one was given).

| Method | FID↓ | KID↓ | Precision↑
Recall↑ | Diversity↑ |
|---|---|---|---|---|
| ACTOR (2021) | 48.80 | 0.53 | 0.72, 0.74 | 14.10 |
| MoDi (2022) | **13.03** | **0.12** | **0.71**, **0.81** | **17.57** |
| MDM (ours) | 31.92 | 0.36 | 0.66, 0.62 | 17.00 |

Table 5: **Evaluation of unconstrained synthesis on the HumanAct12 dataset.** We test MDM in the challenging unconstrained setting, and compare with MoDi (Raab et al., 2022), a work that was specially designed for such setting. We demonstrate that in addition to being able to support any condition, we can achieve plausible results in the unconstrained setting. **Bold** indicates best result.

## 5.2 UNCONSTRAINED SYNTHESIS

The challenging task of unconstrained synthesis has been studied by only a few (Holden et al., 2016; Raab et al., 2022). In the presence of data labeling, e.g., action classes or text description, the labels work as a supervising factor, and facilitate a structured latent space for the training network. The lack of labeling make training more difficult. The human motion field possesses rich unlabeled datasets (Adobe Systems Inc., 2021), and the ability to train on top of them is an advantage. Daring to test MDM in the challenging unconstrained setting, we follow MoDi(Raab et al., 2022) for evaluation. We use the metrics they suggest (FID, KID, precision/recall and multimodality), and run on an unconstrained version of the HumanAct12 (Guo et al., 2020) dataset.

**Data.** Although annotated, we use HumanAct12 (see Section 4.2) in an unconstrained fashion, ignoring its labels. The choice of HumanAct12 rather than a dataset with no labels (e.g., Mixamo (Adobe Systems Inc., 2021)), is for compatibility with previous publications.

**Implementation.** Our model uses the same architecture for all forms of conditioning, as well as for the unconstrained setting. The only change to the structure shown in Figure 2, is the removal of the conditional input, such that $z_{tk}$ is composed of the projection of $t$ only. To simulate an unconstrained behavior, ACTOR Petrovich et al. (2021) has been trained by (Raab et al., 2022) with a labeling of one class to all motions.

**Quantitative evaluation.** The results of our evaluation are shown in table 5. We demonstrate superiority over works that were not designed for an unconstrained setting, and get closer to MoDi (Raab et al., 2022). MoDi is carefully molded for unconstrained settings, while our work can be applied to any (or no) constrain, and also provides editing capabilities.

## 6 DISCUSSION

We have presented MDM, a method that lends itself to various human motion generation tasks. MDM is an untypical classifier-free diffusion model, featuring a transformer-encoder backbone, and predicting the signal, rather than the noise. This yields both a lightweight model, that is unburdening to train, and an accurate one, gaining much from the applicable geometric losses. Our experiments show superiority in conditioned generation, but also that this approach is not very sensitive to the choice of architecture. A notable limitation of the diffusion approach is the long inference time, requiring about 1000 forward passes for a single result. Since our motion model is small anyway, using dimensions order of magnitude smaller than images, our inference time shifts from less than a second to only about a minute, which is an acceptable compromise. As diffusion models continue to evolve, besides better compute, in the future we would be interested in seeing how to incorporate better control into the generation process and widen the options for applications even further.

ACKNOWLEDGEMENTS

We thank Rinon Gal for his useful suggestions and references. This research was supported in part by the Israel Science Foundation (grants no. 2492/20 and 3441/21), Len Blavatnik and the Blavatnik family foundation, and The Tel Aviv University Innovation Laboratories (TILabs).

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

# A  ADDITIONAL EXPERIMENTS

## A.1  DIFFUSION PARAMETERS

Learning MDM with different numbers of diffusion steps significantly affects performance and holds the potential to accelerate inference time. Table 6 shows optimal performance for $T = 100$, in addition, it enables accelerating inference by a factor of 10 compared to $T = 1000$, which is widely used for images.

| Diffusion steps ($T$) | R Precision (top 3)↑ | FID↓ | Multimodal Dist↓ | Diversity→ |
|---|---|---|---|---|
| Real | $0.797^{\pm.002}$ | $0.002^{\pm.000}$ | $2.974^{\pm.008}$ | $9.503^{\pm.065}$ |
| 10 | $0.574^{\pm.006}$ | $1.461^{\pm.088}$ | $5.816^{\pm.033}$ | $\underline{9.369^{\pm.058}}$ |
| 100 | $\underline{0.640^{\pm.007}}$ | $\mathbf{0.454^{\pm.039}}$ | $\underline{5.336^{\pm.029}}$ | $9.906^{\pm.053}$ |
| 500 | $\mathbf{0.662^{\pm.007}}$ | $0.553^{\pm.055}$ | $\mathbf{5.177^{\pm.028}}$ | $9.890^{\pm.074}$ |
| 1000 | $0.611^{\pm.007}$ | $\underline{0.544^{\pm.044}}$ | $5.566^{\pm.027}$ | $\mathbf{9.559^{\pm.086}}$ |

Table 6: **Diffusion steps (HumanML3D test set).** We run all the evaluation 20 times. **Bold** indicates best result, underline indicates second best, $\pm$ indicates 95% confidence interval, $\rightarrow$ indicates that closer to real is better.

## A.2  GEOMETRIC LOSSES

We conduct a thorough experiment to evaluate the contribution of geometric losses with the Human-Act12 dataset. The results are presented in Table 7. For alignment with prior work, all metrics are calculated using the deep features of the action recognition network suggested by Guo et al. (2020). In general, MDM scores are too close to the real test distribution (i.e. the evaluation network fails to discriminate between the two). This means that quantitative results comparing the different variants MDM are too similar to evaluate. As a result, we are not able to decide what combination of geometric losses is preferred. We leave for future work experimenting with a different, more expressive, evaluation network.

| Method | FID↓ | Accuracy↑ | Diversity→ | Multimodality→ |
|---|---|---|---|---|
| Real | $0.050^{\pm.000}$ | $0.990^{\pm.000}$ | $6.880^{\pm.020}$ | $2.590^{\pm.010}$ |
| MDM (ours) | $0.100^{\pm.000}$ | $\mathbf{0.990^{\pm.000}}$ | $\mathbf{6.860^{\pm.050}}$ | $2.520^{\pm.010}$ |
| w/o foot contact | $\mathbf{0.080^{\pm.000}}$ | $\mathbf{0.990^{\pm.000}}$ | $6.810^{\pm.010}$ | $\underline{2.580^{\pm.010}}$ |
| w/o geometric losses | $\underline{0.090^{\pm.000}}$ | $\mathbf{0.990^{\pm.000}}$ | $6.820^{\pm.020}$ | $2.550^{\pm.010}$ |
| foot contact only | $0.100^{\pm.000}$ | $\mathbf{0.990^{\pm.000}}$ | $\mathbf{6.860^{\pm.050}}$ | $2.520^{\pm.010}$ |
| velocity only | $0.100^{\pm.000}$ | $\mathbf{0.990^{\pm.000}}$ | $6.820^{\pm.020}$ | $\mathbf{2.590^{\pm.000}}$ |
| pose only | $\underline{0.090^{\pm.000}}$ | $\mathbf{0.990^{\pm.000}}$ | $\underline{6.830^{\pm.020}}$ | $2.570^{\pm.020}$ |

Table 7: **Geometric losses ablation study.** (HumanAct12 dataset) the relative $\lambda$ equals 1 when the loss term is included, and 0 when it is excluded.

# B    EVALUATION METRICS.

For the completeness of our work, we describe here the quantitative metrics used throughout the paper, as they originally described and implemented by Guo et al. (2020) for action-to-motion and by Guo et al. (2022a) for text-to-motion.

## B.1    ACTION-TO-MOTION

The following metrics are based on an RNN action recognition network as it was originally trained by Guo et al. (2020). We refer to it as the evaluator network.

**Frechet Inception Distance** (FID). A widely used metric to evaluate the overall quality for generation tasks. FID is calculated upon features extracted from 1,000 generated motion vs ground truth (real) taken from the test set. To adjust this metric to the motion domain, we extract a deep representation of the motion with the evaluator network instead of the inception neural network, originally used for images. A lower value implies better FID results.

**Accuracy.** We classify 1,000 generated motions using the evaluator network, than we calculate the overall recognition accuracy that indicates the correlation of the motion and its action type.

**Diversity** measures the variance of the generated motions across all action categories. We first randomly sample two subsets of the same size $S_d$ out of a set of all generated motions across all action categories denoted $\{\mathbf{v}_1, ..., \mathbf{v}_{S_d}\}$ and $\{\mathbf{v}_1'', ..., \mathbf{v}_{S_d}'\}$. The diversity of those sets of motions is defied as

$$\text{Diversity} = \frac{1}{S_d} \sum_{i=1}^{S_d} \| \mathbf{v}_i - \mathbf{v}_i' \|_2 . \tag{6}$$

We use $S_d = 200$ for our experiments. The diversity value is considered better when closer to the diversity value of the ground truth.

**Multimodality** measures the generated motions diversify within each action class. We randomly sample two subsets with size $S_l$ of the same motion class $c$ $\{\mathbf{v}_{c,1}, ... , \mathbf{v}_{c,S_l}\}$ and $\{\mathbf{v}_{c,1}', ..., \mathbf{v}_{c,S_l}'\}$. The multimodality of all action classes $C$ is defined as,

$$\text{Multimodality} = \frac{1}{C \times S_l} \sum_{c=1}^{C} \sum_{i=1}^{S_l} \left\| \mathbf{v}_{c,i} - \mathbf{v}_{c,i}' \right\|_2 . \tag{7}$$

We use $S_l = 20$ for our experiments.

## B.2    TEXT-TO-MOTION

Originally suggested by Guo et al. (2022a), the following metrics are based on a text feature extractor and motion feature extractor jointly trained under contrastive loss to produce geometrically close feature vectors for matched text-motion pairs, and vise versa.

**R Precision. (top-3)** For each generated motion, its ground-truth text and a randomly selected miss-matched descriptions from the test set. We calculate the euclidean distance between the motion feature and text feature of each description in the pool. We count the average accuracy at top-3 places. If the ground truth entry falling into the top-3 candidates, we treat it as True Positive retrieval. We use a batch size 32 (i.e. 31 negative examples).

**FID.** Same as for action-to-motion, using the motion extractor as the evaluator network.

**Multimodal Distance.** We calculate the multimodal distance as the average Euclidean distance between the motion feature of each generated motion and the text feature of its corresponding description in test set. A lower value implies better multimodal distance.

**Diversity.** Same as for action-to-motion but with $S_d = 300$.

**Multimodality.** Same as for action-to-motion but with $S_m = 10$.

## C  IMPLEMENTATION DETAILS

The full implementation of MDM can be found in our published code[2]. In addition, the followings are the hyperparameters and model details for all of our experiments.

**Diffusion framework.** In all of our experiments, we used an implementation of DDPM (Ho et al., 2020) by Dhariwal & Nichol (2021)[3]. We use $T = 1,000$ diffusion steps, *cosine* noise scheduling (predefined sigmas). All other hyperparameters are according to the implementation defaults.

**Transformer architecture.** For our transformer architectures, we used the PyTorch implementation[4]. We used 8 transformer layers, 4 attention heads, latent dimension $d = 512$, dropout 0.1, feed-forward size 1024 and *gelu* activations. The number of learned parameters for each model is stated in Table 8.

**GRU architecture.** We use the PyTorch implementation of GRU (Cho et al., 2014)[5] with two layers and latent dimension 512. The number of learned parameters for each model is stated in Table 8.

**Learning hyperparameters.** For all of our experiments, we use batch size 64, learning rate $10^{-4}$.

| Architecture | # Parameters ($\cdot 10^6$) |
|---|---|
| Transformer Encoder | 17.88 |
| Transformer Decoder | 26.29 |
| + input token | 26.29 |
| U-net | 23.47 |
| GRU | 4.47 |

Table 8: The number of learned parameters per architecture for the text-to-motion task. For the action-to-motion task, there are additional 512 parameters per-class for the class embeddings module.

## D  USER STUDY

In Section 4.1 we conduct a user study for the text-to-motion task. We asked 31 users to choose between MDM and state-of-the-art works in a side-by-side view, with both samples generated from the same text prompt randomly sampled from the KIT test set. We repeated this process with 10 samples per model and 10 repetitions per sample. This user study enabled a comparison with the recent TEMOS model (Petrovich et al., 2022), which was not included in the HumanML3D benchmark. Fig. 4 shows that most of the time, MDM was preferred over the compared models, and even preferred over ground truth samples in 42.3% of the cases. This user study was designed to measure the precision of the models, i.e. which one better fits the input text. The exact phrasing of the question was "Which animation better fits the following description?". A sample question from this study is presented in Fig. 5.

---

[2]https://github.com/GuyTevet/motion-diffusion-model
[3]https://github.com/openai/guided-diffusion
[4]https://pytorch.org/docs/stable/nn.html
[5]https://pytorch.org/docs/stable/generated/torch.nn.GRU.html

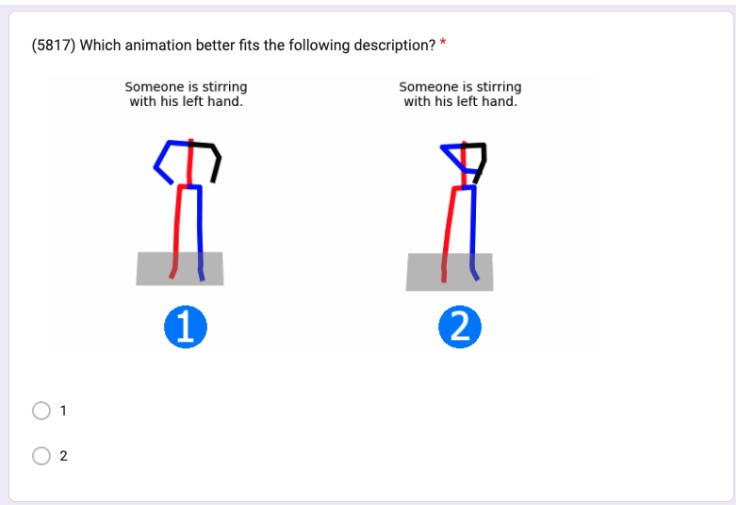

Figure 5: An example question for our text-to-motion user study, using the Google Forms platform.

