# OpenReview forum: "Human Motion Diffusion Model"
_ICLR.cc/2023/Conference — ICLR 2023 notable top 25%_

### Official Review · Reviewer_Eywo · 2022-10-23

**Confidence:** 4
**Correctness:** 4
**Technical Novelty And Significance:** 3
**Empirical Novelty And Significance:** 3
**Recommendation:** 8

**Clarity, Quality, Novelty And Reproducibility:**

The paper is clear, and the presentation and results are of high quality. The novelty of the paper seems to be there in mixing transformer backbones with a diffusion process. I don't think there is enough detail to reproduce the work. Things such as number of attention heads, hidden dimension, and other details of the transformer backbone are missing.

**Strength And Weaknesses:**

Strengths:
+ New diffusion architecture that combines transformers and diffusion
+ Quantitatively and Qualitatively outperforms baselines in most of the evaluation tasks.
+ Nice visualizations of the diffusion process and results


Weaknesses:

- Comparison between simply U-Net and the proposed backbone.
Given that the architectural novelty is the use of transformers instead of U-Net, I suggest the authors provide an ablation comparing the two design choices. The ablation should compare performance, inference time, and maybe any analysis showing differences of one versus the other. I feel like this is something critical that is missing from the paper, and would be good to have.


- Inference time?
Given that diffusion models are known for being slow, it would be good to provide the time it takes for inference in comparison to other models. This would show how much room for improvement there is in this aspect of the method, and the tradeoffs of using the proposed method versus the performance gain over the baselines.

**Summary Of The Paper:**

This paper proposes a diffusion based motion modeling method that uses transformers a backbone for the diffusion process. The authors train a general conditional motion generation method using classifier free guidance which allows them to use the same model for conditioned and unconditioned generation. In experiments, the authors perform favorably against other motion generation methods in the tasks for unconditional generation, text-conditioned generation, action-conditioned generation, in-betweening, and motion editing.

**Summary Of The Review:**

All-in-all, this is a nice paper with good results and nice presentation. However, I am slightly learning towards acceptance due to the concerns presented in the weakness section, and lack of detail for reproducibility. Nevertheless, I am looking forward to the author's response and am willing to increase my score depending on it.

---

> ### Author Response · Authors · 2022-11-18
> **Reply to reviewer Eywo**
>
> Thank you for your thoughtful comments.
>
> **“Comparison between simply U-Net and the proposed backbone…”**
> Following your comment, we adapt U-net for motion data and train it with the HumanML3D benchmark, as the other backbone ablations, and report the results in the paper. We show that it is possible to learn motion with U-Net but with inferior performance compared to the transformer architecture. In addition, we find transformer architecture lighter in terms of memory consumption and runtime.
>
> **“Inference time?...”**
> As mentioned in the discussion section, inference time for diffusion models is indeed a limitation, although in this case, the small dimensionality of the motion tensor allows inference in less than a minute. Furthermore, we tried using DDIM sampling to accelerate the process. DDIM yields motion with a reasonable appearance, yet significantly higher FID, so we decided to leave this experiment out of the scope of the paper. Another more fruitful attempt is to learn MDM with fewer diffusion steps: Following your question, we added to the appendix an experiment measuring FID as a function of T. The results suggest that T=100 achieves already comparable results to the default T=1000, and holds the potential to accelerate sampling by a factor of 10.
>
> **“Things such as number of attention heads, hidden dimension, and other details of the transformer backbone are missing…”**
> Added to the appendix following your suggestion. Thanks.

---

> > ### Comment · Reviewer_Eywo · 2022-11-28
> > **Response to authors'**
> >
> > I would like to thank the authors for addressing my concerns. I have increase my score.

---

> > > ### Author Response · Authors · 2022-11-28
> > > **Reply to Eywo**
> > >
> > > Thank you!

---

### Official Review · Reviewer_W5bV · 2022-10-24

**Confidence:** 5
**Correctness:** 3
**Technical Novelty And Significance:** 3
**Empirical Novelty And Significance:** 4
**Recommendation:** 8

**Clarity, Quality, Novelty And Reproducibility:**

The paper is generally well written despite a few missing details (mentioned above), and I think the quality of the work is high considering the principled design taken for the proposed diffusion model and extensive experiments. As mentioned above, the work is novel in how it combines previous diffusion components for human motion and I think it will inspire many followup works.

Between the paper details and code (promised to be released), the paper is reproducible.


**Strength And Weaknesses:**


Strengths:
* The adaptation of the diffusion framework is done in a principled way, with reasonable design choices suitable for human motion modeling. For example, predicting the final clean motion from the network instead of the noise at each step is a subtle but important choice, which allows using geometry penalties during training and easy editing through overwriting at test time. Also using classifier-free guidance to control the influence of conditioning. Though no component is particularly novel (i.e. they are adapted from other domains), their combination gives a strong first pass at diffusion for human motion.
* The approach is novel in the sense that it is the first diffusion model for full body motion (besides concurrent arXiv papers which show diffusion being used in a more limited capacity).
* The model is demonstrated on a wide range of tasks (text-conditioned, action-conditioned, unconditioned, in-filling, editing). This large task variation really shows the capabilities of diffusion for human motion which may push the community to creatively continue in this direction.
* Quantitively, the model is competitive or better than several relevant baselines across the evaluated tasks. Qualitatively, the results are quite convincing. Overall, the evaluation is quite thorough.
* Code to be published.

Weaknesses:
* A couple components of the architecture were not clearly justified. First, the use of a transformer rather than the usual U-Net. The intro says the transformer “better fits the temporal and non-spatial nature of motion data” but a U-Net baseline is not compared against in experiments.  Second, why is CLIP is used rather than some other language-only model (as in TEMOS) to get the text embedding? Is the “visual” nature of the CLIP latent space actually helpful for human motion?
* Sec 3 could use a bit more detailed intro to diffusion models for those not familiar, especially since this model is new to human motion.  For example, before Eq (2) it is stated that the model predicts the clean samples rather than the noise, but doesn’t explain that these are equivalent formulations.
* Eq. 7 combines the clean motion predictions from the network for classifier-free guidance. This is a bit different from the original formulation of Ho & Salimans, which combines the noise prediction from the model. Are these formulations equivalent? (perhaps this doesn’t matter in practice).
* For the user study in Sec 4.1. it’s not clear what the users were asked to evaluate. Is it the motion quality? Is it how well the motion matches the text? Ideally both should be evaluated.
* Sampling from the diffusion model is quite slow (~1min for a sequence) as discussed in the conclusion. However, there has been much recent work on speeding up sampling lately, so this can be improved in future work.

Questions, comments, and clarifications (not considered in rating):
* How many parameters are in the architecture? Is the transformer considered “lightweight” because it is faster or smaller than U-Net?
* Intro: saying “non-spatial nature of motion data” is a bit confusing. Joints are positions which are by definition “spatial”. Maybe the authors meant the data is less structured, but even so joints are part of the kinematic tree structure.
* Related work: Gu et al [Stochastic Trajectory Prediction via Motion Indeterminacy Diffusion, CVPR 2022] use a similar diffusion transformer for pedestrian trajectory prediction.
* In practice, how long can generated sequences be before motion quality starts to degrade? Or can it truly handle “arbitrary” lengths?
* If both joint positions and rotations are available in the pose, which are actually used as the output?
* Since there are so many evalution metrics, it would be good to define them in the appendix for easy access.
* It would be interesting to see how classifier guidance works for text and action-to-motion instead of having to re-train separately for each task.


**Summary Of The Paper:**

This paper introduces a denoising diffusion model for both conditional and unconditional human motion (pose sequence) generation. Except for a couple of concurrent arXiv papers, this is the first time diffusion models have been applied to full-body human motion. The paper proposes a transformer-based model that can be conditioned on a text or action embedding in order to synthesize specific motions. The proposed approach makes several design decisions that make the diffusion model amenable to human motion including predicting clean samples rather than noise at each step and applying additional losses to improve motion quality. The model is evaluated on a large variety of tasks including text/action-conditioned generation, unconditioned generation, in-betweening, and motion editing, showing strong performance both qualitatively and quantitatively.

**Summary Of The Review:**

Overall, the paper clearly introduces and thoroughly evaluates a strong diffusion model for human motion. Despite some small weaknesses, I think the qualitative results and flexibility of the model make it very useful for the community. Therefore, I recommend accept.

---

> ### Author Response · Authors · 2022-11-18
> **Reply to reviewer W5bV**
>
> Thank you for your thoughtful comments.
>
> **“First, the use of a transformer rather than the usual U-Net…”**
> Following your comment, we adapt U-net for motion data and train it with the HumanML3D benchmark, as the other backbone ablations, and report the results in the paper. We show that it is possible to learn motion with U-Net but with inferior performance compared to the transformer architecture. In addition, we find transformer architecture lighter in terms of memory consumption and runtime.
>
> **“Second, why is CLIP is used rather than some other language-only model (as in TEMOS) to get the text embedding…”**
> Our text-to-motion and action-to-motion experiments are demonstrating the ability of MDM to handle an arbitrary control signal. As a result, we chose to encode action/text using general baseline approaches rather than encoders that are optimized for the problem. Following your question, we switch CLIP with the sentence-BERT model. The results indicate comparable performance, indicating that MDM is not sensitive to the choice of the encoder.
>
> **Eq 2** - added clarification for the equivalency of predicting the noise and the signal.
>
> **Eq 7** - This is a fine observation. Indeed, Ho & Salimans suggested classifier-free sampling for epsilon prediction. This paper shows empirically that classifier-free sampling works also when predicting the signal rather than the noise.
>
> **“For the user study in Sec 4.1…”**
> Our user study asks “Which animation better fits the following description?”, following your question we clarified this in our writing and added a sample frame from the user study to avoid any ambiguity.
>
> **“Sampling from the diffusion model…”**
> In the discussion section, we noted inference time for diffusion models as a limitation, although the small dimensionality of the motion tensor allows inference in less than a minute. Furthermore, we tried using DDIM sampling to accelerate the process. DDIM yields motion with reasonable appearance, yet significantly increases FID, hence we decided to leave this experiment out of the scope of the paper. Another more fruitful attempt is to learn MDM with fewer diffusion steps: Following your question, we added to the appendix an experiment measuring FID as a function of T. The results suggest that T=100 achieves already comparable results to the default T=1000, and holds the potential to accelerate sampling by a factor of 10.
>
> **“How many parameters are in the architecture”**
> Thanks for this question. The full details were added to the appendix.
>
> **“Intro: saying “non-spatial nature of motion data” is a bit confusing…”**
> Thanks, we rephrased the sentence.
>
> **“Related work: Gu et al…” **
> Thanks, the work was added.
>
> **“In practice, how long can generated sequences”**
> Good question. In our experiments, we tested sequence lengths according to the test set distribution (up to 10 seconds for the HumanML3D dataset). In the future, it will be interesting to qualitatively check if MDM is capable of generating longer sequences.
>
> **“If both joint positions and rotations are available in the pose, which are actually used as the output?”**
> In the datasets HumanAct12 and UESTC, The data is represented by SMPL joints, which are directly learned and predicted by MDM. In the datasets HumanML3D and KIT the motion is represented by both positions and rotations as you mentioned. For evaluation, both are used to calculate FID and the other quantitative metrics. The rotations in this data are not completely aligned with the SMPL convention. Hence, for visualization, if only a skeleton is needed - positions are used. For SMPL mesh visualization we follow HumanML3D author guidelines and run the SMPLify algorithm on the output positions, which outputs SMPL rotations.
>
> **“Since there are so many evalution metrics,...”**
> Following your comment, we added a section defining them in the appendix.
>
> **“It would be interesting to see how classifier guidance…”**
> We did not understand this comment. Can you clarify?

---

> > ### Comment · Reviewer_W5bV · 2022-11-22
> > **re: reply**
> >
> > Thank you for the replies and the new experiments verifying that a transformer is indeed better than the U-Net and that MDM still works with other language encoders. This has addressed my concerns and I will keep my initial rating of accept.

---

### Official Review · Reviewer_Cmzb · 2022-10-25

**Confidence:** 4
**Correctness:** 4
**Technical Novelty And Significance:** 3
**Empirical Novelty And Significance:** 4
**Recommendation:** 8

**Clarity, Quality, Novelty And Reproducibility:**

The paper is clearly written and has relatively high quality. The presented work appears original, although its arxiv version can be easily located via a search engine, I tend to believe the submission is not a duplicate work if the author lists are the same.

**Strength And Weaknesses:**

Paper strengths:
- The paper presents a novel method, which is among the first to apply diffusion models to human motion synthesis. The method is well-thought, containing multiple non-trivial design choices to make the method work well on human motion data.
- The paper is well-written and easy to follow. The demo is high-quality and well-made.
- The experimental evaluations are thorough and convincingly demonstrate the effectiveness of the proposed model.

Paper weaknesses:
- Intuitively, the synthesized human motion should be varying-length. However, this does not seem to be addressed in the paper. I assume the clip length was fixed and sequences were temporally stretched/squeezed to meet the specified length. It would be great if additional discussions can be added.
- It would be great to see a more detailed analysis of each particular geometric loss term and the influence of its weighting.

**Summary Of The Paper:**

The paper presents a diffusion model for synthesizing human motion parameters. The proposed model consists of a diffusion model adapted to the human motion data format, plus task-specific geometric losses. The proposed model is applied to multiple scenarios such as text-to-motion, action-to-motion, motion editing, and unconstrained synthesis. Extensive quantitative and qualitative evaluations show the promising performance gain achieved by the proposed model.

**Summary Of The Review:**

Overall I think this is a strong paper, which presents the first steps of applying diffusion models to human motion synthesis. The presented method is technically sound and contains multiple insightful details. The paper's evaluation is comprehensive and its demonstration is well-executed. My concerns are minor and can be easily addressed. Therefore, my initial recommendation is to accept the submission.

---

> ### Author Response · Authors · 2022-11-18
> **Reply to reviewer Cmzb**
>
> Thank you for your thoughtful comments.
>
> **“Intuitively, the synthesized human motion should be varying-length…”**
> Varying-length motions are supported by MDM and results for this setting are presented for the KIT and HumanML3D benchmarks. Thanks for this comment, we clarify this point in our writing.
>
> **“It would be great to see a more detailed analysis of each particular geometric loss…”**
> Following your comment, we added a thorough analysis in the appendix.

---

> > ### Comment · Reviewer_Cmzb · 2022-11-21
> > **Response to rebuttal**
> >
> > I appreciate the authors' additional efforts to improve the paper. My main concerns have been addressed successfully. I am maintaining my initial rating of 8.

---

### Official Review · Reviewer_Lxvz · 2022-10-25

**Confidence:** 4
**Correctness:** 3
**Technical Novelty And Significance:** 2
**Empirical Novelty And Significance:** 3
**Recommendation:** 6

**Clarity, Quality, Novelty And Reproducibility:**

The writing is clear and easy to understand, but the motivation for using diffusion model (predicts signal) is not clear yet.

The quality is good shown in the demos. However, some ablation experiments are missing (stated above), which is important.

The proposed method is somewhat novel and the code will be published.

**Strength And Weaknesses:**

Pros
- A simple and general framework, along with some good-quality generation results. "simple" is for the model simplicity, and "general" is for the ability to enable different forms of condition (text, action, editing...)
- Lightweight compared to other multi-modality motion generation works.
- Easy to read and understand.
- Rich demos.

Cons

The necessity of the diffusion model is not clearly presented, except for adding geometric losses. The motivation for using diffusion models which predict the signal itself instead of the noise still needs a discussion. If the reason is that geometric losses could be added, given that we supervise each MDM with the GT $x_0$, it seems that there is no need to stack MDM for the diffusing process.

- What will the result be when $T=1 or  2$ both for training and testing, compared to $T=1000$?  If there is just a small difference, it seems that the good performance is contributed to the transfomer encoder and CLIP, which has been widely used in many works.

- Another concern is about the low diversity caused by predicting the signal at each step, which seems opposite to the original goal of diffusion models.

- Have the authors tried to predict $\epsilon$ and not use geometric losses in MDM?

Ablation experiments on diffusion models are appreciated and needed since they could help readers understand the motivation for introducing the diffusion model better.



**Summary Of The Paper:**

This paper proposes the Motion Diffusion Model (MDM) for multi-modality human motion generation, such as text-to-motion, action-to-motion, and unconditioned generation, etc. MDM is transformer-based and adopts an existing diffusion method that predicts the signal itself at each timestep instead of the noise so that geometric losses could be used. MDM achieves impressive results on several motion generation tasks, and requires lower computation resources at the same time.

**Summary Of The Review:**

Overall, I lean to accept this paper at this time, since it is a simple and general framework with promising results in several motion generation tasks. However, I'm still confused about the necessity of introducing diffusion here, especially in predicting the signal itself. My rating may change depending on the experiment results I mentioned above.

---

> ### Author Response · Authors · 2022-11-18
> **Reply to reviewer Lxvz**
>
> Thank you for your thoughtful comments.
>
> **“What will the result be when T=1or2 both for training and testing…”**
> Following your suggestion, we added an experiment presenting FID and precision as a function of T. The results are pretty interesting - it seems that reducing the number of diffusion steps, even down to T=100, slightly improves results. It suggests that in the case of motion data, lesser diffusion steps are required, and the sampling process can be accelerated as a result. For very low T values (converging back to an autoencoder approach essentially), we report dramatic performance drops, as can be expected.
>
> **“Another concern is about the low diversity caused by predicting the signal…”**
> We find that MDM generates a large diversity of motions given a condition. We demonstrate that result with (1) SOTA results for the diversity metric across different benchmarks, and (2) the supplementary video qualitatively demonstrates the diversity of motions generated for two text prompts.
>
> **“Have the authors tried to predict ϵ and not use geometric losses in MDM?”**
> This experiment unfortunately diverges. Following your question, we report that result in the paper for the completeness of the work.
>
> **“Ablation experiments on diffusion models are appreciated and needed”**
> We agree, hence adding a report ablation for T value and training to predict epsilon. If you have any other suggestions, please share them with us.

---

> > ### Comment · Reviewer_Lxvz · 2022-11-21
> > **Response to authors**
> >
> > Thanks for adding additional experimental results. They help me to better understand the method. Although the experiments still could not fully solve my questions, given the general framework, completeness and good results, my rating still keeps the same.
> >
> > The point that I'm still confused about is why predicting $\epsilon$ fails. I notice another concurrent work named MotionDiffuse [1] doing the same task, text-to-motion, which also has a Transformer architecture and predicts $\epsilon$ and gets promising results. I'm seeking some insights into why in MDM, predicting $\epsilon$ would fail.
> >
> > [1] MotionDiffuse: Text-Driven Human Motion Generation with Diffusion Model

---

> > > ### Author Response · Authors · 2022-11-21
> > > **Reply regarding epsilon prediction**
> > >
> > > Thank you for this comment. Indeed you mentioned a concurrent work (which is cited in our paper) which do predict epsilon.
> > > Please note that their model is trained on 8 V100 GPUs with batch size 1024, whereas our model was trained on a single GPU with batch size 64. Observing that, our conjecture is that a very large batch size is required for learning epsilon, although we don't have the resources to validate it. Anyway, judging the quality of the results, we reckon that predicting X_0 with geometric losses is favorable, even regardless of the cheaper resources needed.

---

> > > > ### Comment · Reviewer_Lxvz · 2022-11-21
> > > > **Response to the epsilon reply**
> > > >
> > > > Thank you for the clarification. Hope to see a deeper exploration of MDM in the future.

---

### Author Response · Authors · 2022-11-18
**General response to reviewers**

We thank the reviewers for the thorough review and thoughtful comments. We are happy to see that you thought MDM is **“a simple and general framework with promising results in several motion generation tasks”**(Lxvz),
That the paper **“clearly introduces and thoroughly evaluates a strong diffusion model for human motion.”** (W5bV), that it is **“a strong paper”**,…**“technically sound and contains multiple insightful details”**, and that **"the paper's evaluation is comprehensive”** (Cmzb).

In the updated version of the paper, changes are marked in blue.
The following summarizes the new experiments added to the paper according to your suggestions:
1. First and foremost, we experiment with different numbers of diffusion steps (T).It turns out T=100 performs slightly better than the default T=1000 while accelerating inference by a factor of 10! Lower values of T impair performance.
2. We experiment with a U-Net variant of MDM with 1D convolutions, to fit motion data. We show that it is possible to learn motion with U-Net but with inferior performance compared to the transformer architecture, both in terms of accuracy, and compute and memory consumption.
3. Training to predict epsilon instead of X_0. This experiment diverges.
4. A comprehensive ablation for geometric losses was added to the appendix. Since MDM performance is too close to real data, all combinations of geometric losses yield similar quantitive performance. Nevertheless, qualitatively, we show that adding geometric losses dramatically improves motion generation (see supplementary video).
5. Replacing CLIP with a BERT-based text encoder. This yields comparable results.

We deeply appreciate your reviews and feel that the new revision brought is significantly more complete thanks to them.
In addition, we address individual comments and questions by commenting on your reviews.

---

### Decision · Program_Chairs · 2023-01-20

**Decision:**

Accept: notable-top-25%

**Justification For Why Not Higher Score:**

Although the authors adequately addressed the concerns brought up in the review along with a revision incorporating some components of feedback, they should be sure to incorporate all the remaining feedback in the final version of the paper.

**Justification For Why Not Lower Score:**

All reviewers are positive on the paper. Considering it's among the first to apply diffusion models to full-body human motion synthesis and will inspire followup work, I recommend accept.

**Metareview: Summary, Strengths And Weaknesses:**

This paper presents a motion diffusion model(MDM) for generating human motion animation given an arbitrary condition or no condition. MDM is a transformer-based approach with light-weight architecture. It adopts an existing diffusion method that predicts the signal itself at each time step instead of the noise so that geometric losses could be used. The model is evaluated on a large variety of tasks both qualitatively and quantitatively.

The strengths include the simple and general framework, the comprehensive evaluation, the promising results, and the well-written content. The main weakness and concerns are some missing details, motivation of using diffusion model, and ablation study. After rebuttal, the authors addressed most of the concerns along with a revision.

**Note From Pc:**

if the above contains the word "oral" or "spotlight" please see: "oral" presentation means -> notable-top-5% and "spotlight" means -> notable-top-25%. As stated in our emails, we are disassociating presentation type from AC recommendations